# Understanding Magnetization Dynamics of a Magnetic Nanoparticle with a Disordered Shell Using Micromagnetic Simulations

**DOI:** 10.3390/nano10061149

**Published:** 2020-06-11

**Authors:** David Aurélio, Jana Vejpravova

**Affiliations:** Department of Condensed Matter Physics, Faculty of Mathematics and Physics, Charles University, Ke Karlovu 5, 121 16 Prague 2, Czech Republic

**Keywords:** magnetic nanoparticles, micromagnetic simulations, magnetization dynamics, hysteresis loop, core-shell structure, spin disorder

## Abstract

Spin disorder effects influence magnetization dynamics and equilibrium magnetic properties of real nanoparticles (NPs). In this work, we use micromagnetic simulations to try to better understand these effects, in particular, on how the magnetization reversal is projected in the character of the hysteresis loops at different temperatures. In our simulation study, we consider a prototype NP adopting a magnetic core-shell model, with magnetically coherent core and somewhat disordered shell, as it is one of the common spin architectures in real NPs. The size of the core is fixed to 5.5 nm in diameter and the shell thickness ranges from 0.5 nm to 3 nm. As a starting point in the simulations, we used typical experimental values obtained for a cobalt ferrite NP of a comparable size investigated previously. The simulations enabled us to study systematically the macrospin dynamics of the prototype NP and to address the interplay between the magnetic anisotropies of the core and the shell, respectively. We also demonstrate how the computational time step, run time, damping parameter, and thermal field influence the simulation results. In agreement with experimental studies, we observed that the direction and magnitude of the shell anisotropy influences the effective magnetic size of the core in the applied magnetic field. We conclude that micromagnetic simulations, in spite of being designed for much larger scales are a useful toolbox for understanding the magnetization processes within a single domain NP with an ordered spin structure in the core and partially disordered spins in the shell.

## 1. Introduction

Research on magnetic nanoparticles (NPs) continues to be viewed with great interest in a wide range of fields like drug delivery [1], bio-analysis [2], data storage [3] and magnetic fluid hyperthermia [4], to name a few. Consequently, better understanding of all fundamental processes behind the coveted magnetic properties is of enormous importance, especially in context of possible applications.

Magnetic NP dynamics, which is in fact the most important physical phenomenon underlying biomedical applications of magnetic NPs [5], can be usually treated within a single domain limit [6]. In experiments, large ensembles of NPs are usually addressed. Therefore mesoscopic factors such as inter-particle interactions, NP size distribution, geometry of the ensemble etc. are superimposed to the single NP behavior. However, understanding the response of a single magnetic NP is of utmost importance, as it has been demonstrated that the internal structure of a single magnetic NP governs the overall response to temperature and magnetic field variations as studied by advanced experimental methods such as Small Angle Neutron Scattering (SANS) and in-field Mössbauer spectroscopy (IFMS) [7,8,9,10,11].

The spin frustration and spin canting effects have been reported as intrinsic to the magnetic NPs [12,13] namely when the NP diameter approaches a few nm size and/or the NPs have a non-negligible structural disorder, which in most cases takes place at the outer part of a NP. Being inspired with reduced saturation magnetization phenomenon in fine magnetic NPs, Coey proposed the famous core-shell model, which considers magnetically aligned core and disordered shell with frustrated spins [14].

Nevertheless, the spin canting and the spin frustration is often discussed in context of the ”surface effects” and the variability of magnetic properties in real NPs with nominally identical chemical composition is often explained via the generic surface anisotropy term. Strictly speaking, the symmetry breaking due to the unsaturated covalent bonds at the surface is given by the symmetry lowering of the coordination environment, which in case of spinel ferrites means that the orbital momentum of the valence states of the transition metal cation in the tetrahedral or octahedral crystal field, is no more quenched and thus contributes to the total magnetic moment of the ion significantly. This scenario assumes that the whole NP is perfect from the crystallographic point of view and only the surface atoms have different magnetic properties due to the symmetry breaking. In real samples, however, the structural arrangement of the atoms in the surface proximity is much more complex and plethora of perturbations already on the level of the lattice, such as point and line defects, local strains, variation of the degree of inversion in the spinel structure, depletion of one of the cations etc., usually take place. The perturbed layer is typically in order of few nanometers thick (depending on the synthesis method), thus these “disorder effects” do have much larger action radius than that of the surface atoms.

Nowadays, the ”core-shell” term gained even a broader context thanks to the growing interest in bimagnetic core-shell NPs, which are considered as more powerful heat generators in the magnetic fluid hyperthermia [15]. The most common architecture of the core-shell NPs is a hard magnetic ferrite in the core (magnetite, cobalt ferrite etc.) and a soft ferrite in the shell (maghemite, manganese ferrite) [16]. Such system can be viewed as a “chemical” core-shell, however, the architecture of such a NP is much more complex from the structural and magnetic point of view.

Focusing back on the ”chemical” core only, it contains already a structurally disordered fraction, most likely at the surface, which brings a kind of “structural” core-shell arrangement. For example in case of spinel iron oxide, the topotactic transition from the maghemite to magnetite makes the structural arrangement even more complex [7,9,12]. The structural disorder including lattice deformation is a base for spin frustration, giving rise to the deviation from the bulk-like coherent magnetic structure. Additional spin canting comes due to intra- or inter-particle interactions [8].

Nevertheless, even more complex scenario can be observed, for example in cobalt ferrite NPs. In case of highly crystalline NPs without internal defects, the spin disorder extends over the entire particle and a homogeneous disordered phase is formed [17]. These observations suggest that the spin disorder at the surface causes a full spin reorganization inside the particle. It seems that the presence of cobalt favors the extended disordered configuration, especially for larger NPs [17], while the magnetically soft ferrites tend to form more variable spin configurations, even though the magnetic core-shell is one of the common ones [7,9,12]. A kind of magnetic core-shell structure originated mostly by the structural disorder has been reported for cobalt ferrite NPs, with size comparable to that treated in our simulations [10,11].

It is obvious that the spin configurations can be rather complex and a simple macrospin approximation often used for description of magnetic NP properties is not sufficient.

In this work we report on an extensive micromagnetic simulation study to model the behavior of magnetic NPs with the magnetic core-shell spin structure observed in many samples of spinel ferrite NPs. We focused on the magnetization isotherms, its inherent coercive field and on how the characteristics of the disordered shell affect the overall NP magnetic behaviour.

In our study, we consider a prototype ${\mathrm{CoFe}}_{2}O_{4}$ NP [11,18], which is composed of two regions: the inner part—the core with completely ordered magnetic (bulk-like) structure and the outer part—the shell with a certain level of spin disorder (see Figure 1). We have to point out that architecture of our model NP corresponds formally to the magnetic core-shell model, however the micromagnetic framework does not allow to account for the origin of the spin disorder in the shell observed in real NPs. Therefore the model can be relevant not only for the pure magnetic core-shell spin arrangements, but when using relevant parameters in the simulations one can apply the same approach for a NP with structural core-shell arrangement, in which the structural disorder in the shell gives rise to the spin disorder.

We also have to point out that the type of magnetic anisotropy in ${\mathrm{CoFe}}_{2}O_{4}$ NPs is not trivial and it is driven by many parameters of the NPs. From the practical point of view, the fabrication process is the driving factor of the NP magnetic properties [19,20,21,22] as it determines the quantity of defects, size and shape (distribution) of the NPs giving rise to the variations of the magnetic properties [23,24]. The ${\mathrm{CoFe}}_{2}O_{4}$ typically present cubic anisotropy in bulk as well as in larger and highly-crystalline NPs [17]. However, different studies reported that in case of small particles (typically with the diameter below 10 nm) a coexistence of both cubic and uniaxial anisotropies is expected and uniaxial anisotropy dominates for NPs with the diameter below ∼5 nm [19,25]. Thus the synthesis processes are capable of changing the dominance of either cubic or uniaxial anisotropy [22]. On top of that, the anisotropy constant of cobalt ferrite NPs can vary significantly with temperature [26]. Given that our prototype NP can be considered as “small”, we focused our micromagnetic simulation studies first on the uniaxial anisotropy, although the influence of ${\mathrm{CoFe}}_{2}O_{4}$ cubic anisotropy is also analyzed and discussed.

Micromagnetic simulations have been successfully used throughout the years [27] to study the magnetization dynamics of magnetic materials at scales much larger than the atomic one but still much smaller than the macroscopic; at the so called microscopic or mesoscopic scale [28]. The magnetization dynamics simulations are based on the Landau-Lifshitz-Gilbert (LLG) equation, where the magnetization of the simulated material is approximated by a continuous vector field with a constant magnitude [29,30,31]. The concept has been extended down to the nanometer scale and it has been routinely applied throughout the years for description of various spintronic devices [32]. Recently, the micromagnetic simulation approach was used to gain more insight into the magnetization dynamics of small NPs within the single domain limit [33,34].

The goal of our study is to demonstrate that micromagnetic simulations can be used to understand the interplay between the ordered spins of the core and disordered spins in the shell in the magnetic core-shell model. We demonstrate here how the magnetic anisotropy and thickness of the shell influence the magnetic properties of such a NP, in particular the character of hysteresis loops at different temperatures. We also discuss importance of realistic adjustment of the simulation parameters, including time step, run time, damping parameter, and thermal field effect. Our study underlines the importance of understanding the magnetization reversal in a single-domain NP beyond the macrospin model.

## 2. Simulation Method

The simulations were performed using the parallel GPU mumax3 [35] finite difference numerical code, where the magnetization dynamics is computed by solving the following LLG equation;
(1)$$\begin{array}{c} \frac{1}{\gamma }\frac{dm}{dt}=-\frac{1}{1+\alpha^{2}}\left(m\times H_{\mathrm{eff}}\right)-\frac{\alpha }{1+\alpha^{2}}m\times \left(m\times H_{\mathrm{eff}}\right) \end{array}$$
where, γ is the gyromagnetic ratio, m=M/$M_{S}$ is the normalized magnetization vector, α is the Gilbert damping parameter and $H_{\mathrm{eff}}$ is the total effective field, which includes the typical fields due to the interactions of exchange, anisotropy, demagnetization as well as the thermal and external fields [30,31,35].

Results of magnetization measurements carried out on ${\mathrm{CoFe}}_{2}O_{4}$ NPs synthesized in our lab were used as a benchmark for the starting magnetic parameters used in the simulations. Thus such parameters as saturation magnetization, $M_{S}$, and anisotropy constant, *K*, are derived from real experimental results, which are in a good agreement of the previously published works, e.g., [16,19]. The parameters are given in Table 1. Figure 1 shows a schematic representation of the core-shell model and Figure 2 shows the evolution of the experimental hysteresis loops at different temperatures. At higher temperatures the superparamagnetic regime is observed, whereas in the blocked state the hysteresis loops open reaching a coercive field of approximately 1.3 T at 10 K. A schematic representation of the spin structure of our model NP is also shown in Figure 2, where one can see the ordered core and the disordered shell spin structure [13].

In agreement with the schematic representation, we modelled our NP with the core-shell structure out of several 0.5 nm side cubic cells in mumax3 [35], where each cell has its own magnetization spin that interacts with all the others dynamically (as can be seen on the NPs snapshot presented in Figure 2). As an example, in the case of the 5.5 nm core 0.5 nm shell thickness NP a total of 1728 computational cells are needed, where about 524 of them correspond to the core magnetization spins and about 381 to the shell. Due to the nature of the finite difference method there is always a staircase effect when building round shapes like spheres, thus the approximation of the number of magnetic cells for the core and shell, as well as the remaining non-magnetic cells of the full computational space.

Considering the experimental data given in Table 1 for our prototype ${\mathrm{CoFe}}_{2}O_{4}$ NP, the parameters shown in Table 2 were defined as the starting set for different simulation runs. As for the remaining computational parameters, a cubic cell size of 0.5 nm side was chosen to build a spherical NP, whose ordered core accounts for 5.5 nm of the inner diameter, whereas the remaining 0.5, 2 and 3 nm account for the partly (or fully) disordered shell of the NP, respectively. For the partly (or fully) disordered shell we make an initial assumption that the magnetic parameters $M_{S}$ and $A_{ex}$ are about 20% smaller than those of the ordered core (Table 2). The reduction of the parameters mimics the variable disorder in real NPs [13]. Please note that the micromagnetic simulations are not capable of including the change in the degree of inversion and the gradients of structural and magnetic parameters on the scale of individual atoms, thus a reasonable approximation in the magnetic parameters must be implemented.

The final response is strongly dependent on the magnetocrystalline anisotropy of the core and the delicate interplay of the exchange interactions between the core and the shell. Therefore we also addressed the effect of core-shell exchange energy, and the anisotropy energy and magnetocrystalline anisotropy direction in the core and in the shell, respectively.

## 3. Results and Discussion

In this section, results of the simulations are presented and discussed within the context of experimental findings. In order to deliver reliable information, we have first investigated how the material-independent parameters like the computational time influence the simulation results, so as to have a base before studying the effects of the material-dependent parameters like, damping, anisotropy, etc.

First, we addressed the role of the time step (FixDt) applied to the numerical solver of the LLG equation, which influences the coercive field ($H_{c}$) of our modelled NP, as does the other analysed time parameter which is the run time (runt). We confirmed that both simulation times influence the magnetization dynamics and, as a rule of thumb, the smaller the time step (in the order of fs) the higher the precision of the calculation. However, there is the drawback that if the time step (FixDt) is too small, the actual time that it takes for the GPU to finish a simulation becomes quite significant, especially if there is a great number of cells describing the system.

The meaning of the run time (runt) parameter is simply how long is the external magnetic field applied, before increasing it by a certain amount, up to the desired maximum value of the applied field when simulating a full hysteresis cycle. The longer the run time is, the more time the magnetic moments have to relax to the ”equilibrium” at that field value.

Finally, the effect of the thermal field, and the influence of the anisotropy and external field directions, respectively, will be discussed.

### 3.1. The Computational Time Step, FixDt

The computational time step (FixDt) is the numerical parameter used within the simulations to advance the time dependence of the LLG differential equation. As it was mentioned before, the smaller it is the higher the precision of the calculation, but at the cost of larger and larger actual simulation times. Results of the magnetization isotherm at 0 K for our above described magnetic NP parameters at different FixDt’s, are shown in Figure 3.

Looking at the results shown in Figure 3 and remembering that a smaller FixDt translate into a more precise calculation, we decided to use a time step no larger than $2\times 10^{-15}$ s, which means a difference of less than 1%, when compared to the coercive field with the smallest FixDt, and half its actual simulation time (Figure 3). This way we were able to have faster simulations than when using the 1 fs time step, while keeping a good precision in the simulated magnetization dynamics.

### 3.2. Run Time per Field Step and the Damping Parameter

The Gilbert damping parameter α, is an Ohmic type dissipation [36] that brings the magnetization back to its equilibrium position, which is given by the effective field $H_{\mathrm{eff}}$ direction, once it has been perturbed. Typical values for α used in micromagnetic thin films vary slightly from 0.01, which is used here as the reference for our simulated NP. On Figure 4 we can see how increasing it shows a minimum for the coercivity of the NP. Being α a phenomenological constant in the LLG equation, it is very difficult to say which value would better represent the experimental sample data shown on Figure 2, specially when considering such a small NP. Since most research is performed in either bulk materials or thin films it is difficult to say which value of α is most appropriate for our NPs. Thus given how influential it can be regarding the value of the coercive field, $H_{c}$, as shown on Figure 4, the remaining simulations were performed assuming typical thin film values for α of 0.01 for the ordered core and 0.005 for the disordered shell of the NP.

Also on Figure 4 we can see the effect of increasing the run time (runt) per field step onto the coercive field. It is clear that the longer each time step of external field is, the lower the coercive field will be, as given by the tested runt’s of $2\times 10^{-11}$ s, $5\times 10^{-11}$ s and $15\times 10^{-11}$ s. This can be understood as the magnetic spins being allowed more time to relax to an equilibrium position at the current $H_{\mathrm{eff}}$, before increasing the field to the next step value during the simulation of the NP’s hysteresis loop.

### 3.3. Thickness of the Disordered Shell and Thermal Field Effect

Figure 5 shows the influence of increasing the disordered shell thickness on the coercive field of the magnetic NP at different temperatures. The temperature effect is introduced in mumax3 [35] as a fluctuating thermal field, according to the description given by Brown [37], which in short introduces a random oscillation term to the LLG effective field that changes at each time step.

Having the disordered shell anisotropy set to zero, we can clearly see in Figure 5 that the thicker the shell is, the bigger the coercive field values, regardless of the temperature set. This result is in coherence with expectations, since if the shell is thicker there is a greater number of magnetic moments to be “switched” as they are forced to follow the core reversal. This effect is ensured by a larger “inertia” to move them away from their position, due to the preferred direction defined by the ordered core $H_{k}$. Even though the anisotropy is zero at the shell, the exchange interaction is still present forcing the magnetic moments to be parallel between nearest neighbour cells, and all the interactions that come from the ordered core (anisotropy, demagnetizing and exchange) are felt by the spins on the disordered shell. In reality our simulations show that the disordered in the shell is continuously reduced with the increasing magnetic field applied to the NP. Consequently, all the spins in both the core and the shell tend to align with the applied magnetic field during the reversal. This behavior is greatly reflected in the experimental observations carried out on cobalt ferrite NPs with a certain level of spin disorder in the shell using field-dependent SANS experiment, which unambiguously reveals the growth of the magnetic volume (extension of the core radius) with increasing magnetic field [11].

We can also ascertain that even a slight increase in temperature significantly reduces the coercivity values comparing to the case at 0 K. As the temperature is further increased the coercivity continues to decrease and above 100 K the transition is a lot less smooth, due to the random oscillations introduced by the thermal field. The reduction of the coercivity is expected when adding the thermal field since this makes it easier to “pull” the magnetization out of its previous equilibrium direction defined by the ordered core anisotropy easy axis, which lies along the x-direction in our simulations.

Figure 6 demonstrates how the increase in temperature affects more strongly the reversal at higher temperatures. Looking there at the insets of the graph at 2 K, the individual spins forming the NP are weekly perturbed by the thermal field during the transition, remaining fairly parallel between nearest neighbouring cells, by their exchange interaction. On the other hand at 300 K, one can see on Figure 6 how the spins now struggle to remain fairly parallel between them, due to the stronger oscillations caused by the thermal field. Complete graphs with insets for all transitions at different temperatures and shell thicknesses can be found on the Appendix A for this work.

Please note that the persistence of coercivity at 300 K is in agreement with the basic formula of a non-interacting macrospin fluctuation, which determines the blocking temperature, T${}_{B}$ (transition temperature between the blocked state and the superparamagnetic state): $\tau_{m}\;=\;\tau_{0}\exp\left(KV\right)$, where $\tau_{m}$ is a characteristic time window of the experiment (which is usually considered in order of 10–100 s for SQUID magnetometry), $\tau_{0}$ is the relaxation time of the macrospin (typically $10^{-9}$ s), *K* is the anisotropy constant (2 $\times \;10^{5}$ J/m${}^{3}$) and *V* is the NP volume. In the prototype NP, the T${}_{B}$ in the thermodynamic equilibrium fall in the 300–450 K temperature interval.

### 3.4. Effect of Anisotropy

Like in other magnetic nanostructures, the type, magnitude and direction of the magnetocrystalline anisotropy play an important role in the magnetization dynamics. This anisotropy interaction is added as a field-type contribution in mumax3 [35]. In this section we will first take a look at the effects of cubic anisotropy, which is intrinsic to the bulk and large highly crystalline NPs of the ${\mathrm{CoFe}}_{2}O_{4}$, and then to the uniaxial one, which is expected to dominate in the NPs of the size comparable to our model structure.

#### 3.4.1. Cubic Anisotropy

While in very small ferrite NPs the uniaxial anisotropy can be considered a good first approximation, ${\mathrm{CoFe}}_{2}O_{4}$ has cubic anisotropy, and although both may be present in real samples, e.g., [19,25,38].

Typical bulk cobalt ferrite NPs have the first and second order anisotropy constant values of $K_{c1}=2.9\times 10^{5}\;J/m^{3}$ and $K_{c1}=4.4\times 10^{5}\;J/m^{3}$, respectively, with easy axis direction (1,0,0). Results of the simulations when applying bulk cubic anisotropy values are shown in Figure 7.

Analysing first the effect of the second order term, $K_{c2}$ (Figure 7a), it is evident that it shows no real change to the magnetization dynamics. This holds true with a thermal field on at 0 K and 300 K.

When comparing the differences between the simulations run with the uniaxial and the cubic anisotropies, with the comparable magnitude of the anisotropy constant (with $H_{\mathrm{ext}}$ along the (1,0,0) direction) the coercivity is found to be about 0.55 T less in the cubic anisotropy case (Figure 7b). However when the external magnetic field is now applied along the y-axis, the coercivity for the cubic anisotropy case is about 0.70 T higher than the uniaxial one (Figure 7c). When considering the thermal field on and with the $H_{\mathrm{ext}}$ applied along the (0,1,0) direction in the uniaxial anisotropy case, the coercivity drops to zero. In the cubic anisotropy case, however, some coercivity plus a minor shift of the hysteresis loop is present, which points to a more complex magnetization dynamics, which is not revealed by the experimental works (e.g., [19]).

#### 3.4.2. Uniaxial Anisotropy

In our simulations the easy axis direction lies along the x-axis of the NP (Figure 1). However, the anisotropy direction was initially defined just for the ordered core of the NP, leaving the anisotropy of the disordered shell as zero, allowing it to follow the core reversal (Table 2).

On Figure 8 we can see how the coercive field varies when the hysteresis field is applied along the anisotropy easy x-axis direction, as well as how it decreases with increasing temperature. More curious is to see what happens when the hysteresis field is now applied perpendicularly to the anisotropy axis (Figure 9). As we can see on Figure 9 the coercivity is significantly reduced when the field is applied perpendicularly to the anisotropy easy axis. This is explained by the fact that applying the external field that way makes it is easier to “pull” the magnetization out of the anisotropy easy direction (x-axis).

It is important to note that the coercivity drops to zero when the thermal field is activated, even for a thermal field as week as the one given by the 2 K temperature. Like it was discussed in the previous sections, the random oscillations introduced by the thermal field help in moving the magnetization out of its equilibrium position. This effect is now amplified by the fact that the external field is now applied perpendicularly to the anisotropy easy axis.

Taking as a reference the 3 nm thick disordered shell simulation shown on Figure 5c let’s explore how the coercivity would change if we defined an easy anisotropy axis for the disordered shell as well, while keeping the temperature at 0 K, so no thermal field contribution is present.

Figure 10 shows the results for the disordered shell with an easy anisotropy direction along the y-axis (0,1,0), while the ordered core remains with its easy direction along the x-axis (1,0,0). This configuration leads to the lower coercivity values at 0 K, when comparing to the original data obtained with no shell anisotropy (Figure 5c). When the magnitude of the anisotropy constant is equal in both the shell and the core, (cross point between the lines in Figure 10) it translates in a large reduction of the coercivity from the case with zero anisotropy in the shell (Figure 5c). Increasing the anisotropy constant for the shell while keeping the core one constant gives a smaller coercivity field, whereas the coercivity increases when the shell anisotropy is reduced. The strong influence of the anisotropy variation in the shell is not that surprising when considering that approximately 73% of the total volume of the NP is now formed by the shell, and thus it greatly affects the overall magnetization dynamics. In this arrangement, it easier to “pull” the overall magnetization from the initial x-direction, since there is a strong interaction arising from all the spins with y-axis as the easy magnetization direction due to the shell anisotropy.

In this vein, this result clearly points to the importance of a correct definition of the NP architecture in micromagnetic simulations, so they can be used to better interpret the underlying physics seen on experimental results.

## 4. Conclusions

In this work, we have performed a micromagnetic simulation study of a prototype NP, which mimics the real internal spin structure observed in many samples of spinel ferrite NPs. In the simulations, we adopted experimental parameters obtained for ∼5–6 nm ${\mathrm{CoFe}}_{2}O_{4}$ NPs. As revealed by the experiments, uniaxial anisotropy dominates in these NPs, however the effect of the cubic anisotropy intrinsic to the bulk material and large NPs has been also tested. We consider that the NP adopts the magnetic core-shell arrangement, which assumes that the model NP is composed of a core represented by a single macrospin as the internal spins are supposed to be aligned in the bulk-like magnetic structure, while the shell contains disordered/paramagnetic-like spin arrangement. Our study reveals that the thickness and anisotropy of the NP shell have a considerable impact on the magnetization reversal, and consequently the shape of hysteresis curves. For example by lowering the anisotropy in the shell, the coherent magnetic volume (corresponding to the effective volume of the core) increases with the applied magnetic field. This result is also in correspondence with recent experimental study carried out on cobalt ferrite NPs [11].

Through our study we demonstrated that micromagnetic simulations are useful tool in reproducing trends in experimental hysteresis curves of single domain NPs and they give reasonable insight into the magnetization dynamics of NPs with the spin structure corresponding to the magnetic core-shell model. However, we also demonstrated here that these simulations are very sensitive to the initial material parameters and adjustments of the simulation procedure. We also note that the micromagnetic concept does not allow to account for the origin of spin disorder; nevertheless, the approach can be applied to pure magnetic core-shell NPs as well to the NPs with the spin disorder induced by the structural disorder of the shell.

Regarding future prospects of our work, we point to the importance of simulation studies on “chemical” core-shell NPs composed of two different materials, containing an interface with a certain level of disorder. Understanding the mechanism of magnetization reversal in these structures is of utmost importance, e.g., for the evaluation of their heating performance in magnetic fluid hyperthermia.

## Figures and Tables

**Figure 1 nanomaterials-10-01149-f001:**
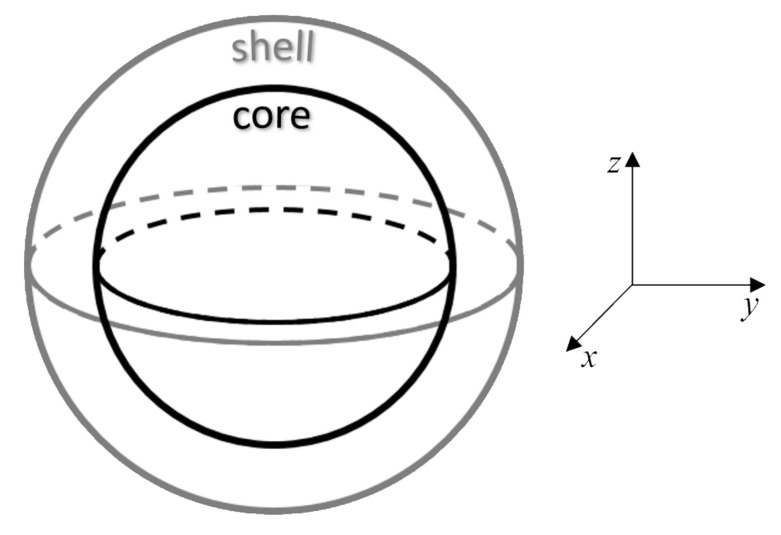
Basic geometry of a magnetic NP used in the micromagnetic simulations showing the real “core-shell” spin structure. The sketch of the coordinate system serves as a guideline for the external magnetic field and anisotropy axis directions used in the simulations.

**Figure 2 nanomaterials-10-01149-f002:**
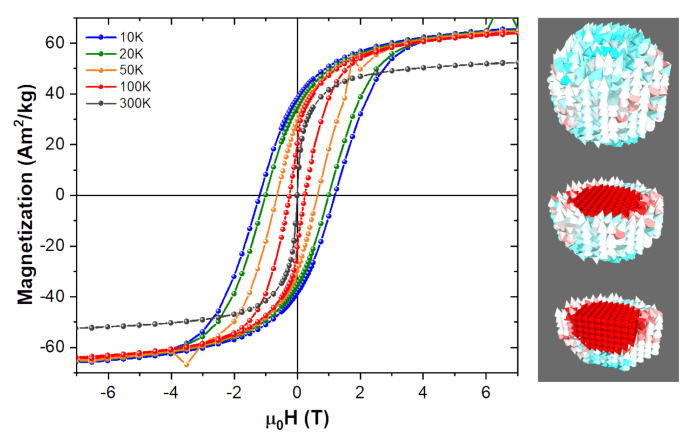
Experimental magnetization isotherms at different temperatures for a real sample of ${\mathrm{CoFe}}_{2}O_{4}$ NPs, as measured on SQUID magnetometer (**left**). A simulated internal spin structure for the model showing both the ordered core and the disordered shell is also shown (**right**).

**Figure 3 nanomaterials-10-01149-f003:**
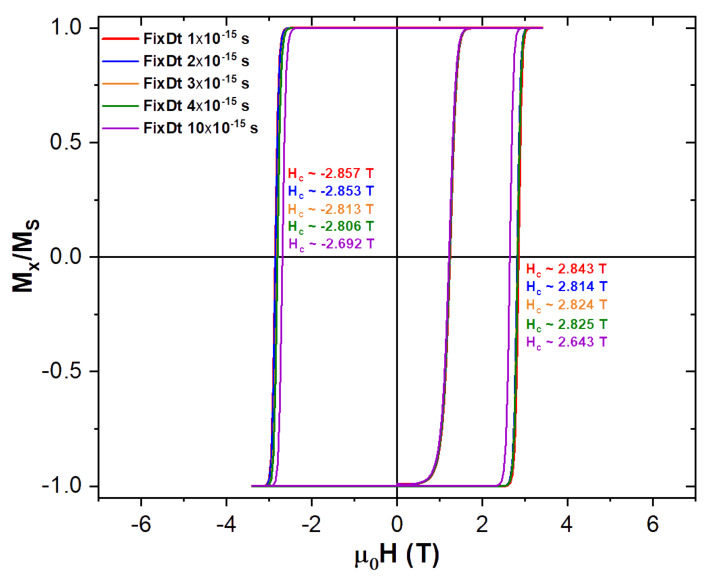
Coercivity ($H_{c}$) variation with the computational time step, FixDt, at 0 K. Smaller FixDt values correspond to a higher numerical precision, but at the cost of significantly longer simulation times. A FixDt of $2\times 10^{-15}$ s was chosen as a good compromise between the precision and simulation times.

**Figure 4 nanomaterials-10-01149-f004:**
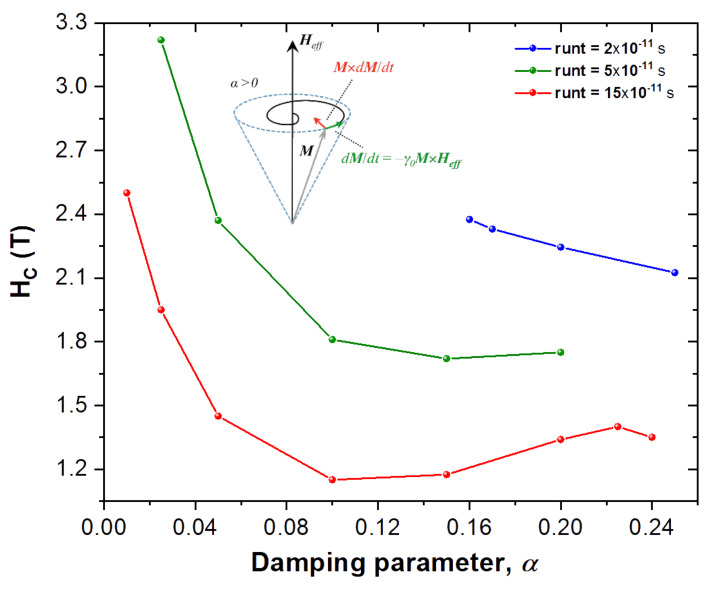
Relation between coercivity ($H_{c}$), damping parameter α, and run time (runt). The inset shows how the dissipative term M×dM/dt, driven by the phenomenological constant α, forces the magnetization to precess until the magnetization aligns itself with the effective field ($H_{\mathrm{eff}}$) direction. In general, as α is increased the coercivity is lowered until a minimum is reached. Also the longer the runt, the longer the magnetization dynamics has to relax to equilibrium at each effective field value, reducing the coercivity as well. These simulations were performed at 0 K.

**Figure 5 nanomaterials-10-01149-f005:**
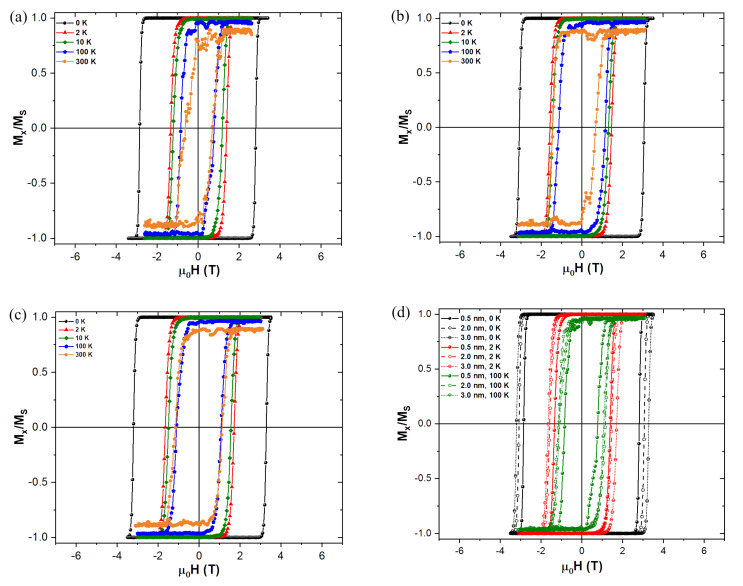
Coercivity changes due to both different simulated temperatures and different disordered shell thicknesses, of 0.5 nm (**a**), 2 nm (**b**) and 3 nm (**c**), whereas (**d**) directly compares the temperature effects for the different shell thicknesses. In general the thicker the shell, the larger the coercivity. As the temperature increases the thermal field effects become more important than the thickness ones, for the overall dynamics of the NP switching.

**Figure 6 nanomaterials-10-01149-f006:**
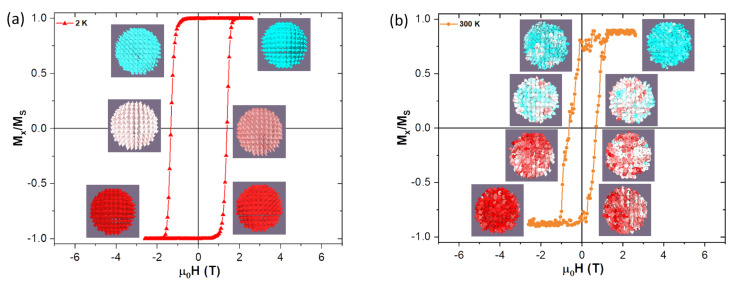
(**a**) Magnetization isotherm reversal for the 0.5 nm shell at 2 K, with corresponding insets showing the spin structure during the transition process. (**b**) Magnetic isotherm reversal for the 0.5 nm shell at 300 K, with corresponding insets showing the spin structure during the transition process. One can easily see on the insets how the individual spins become quite “disorganized” at higher temperatures.

**Figure 7 nanomaterials-10-01149-f007:**
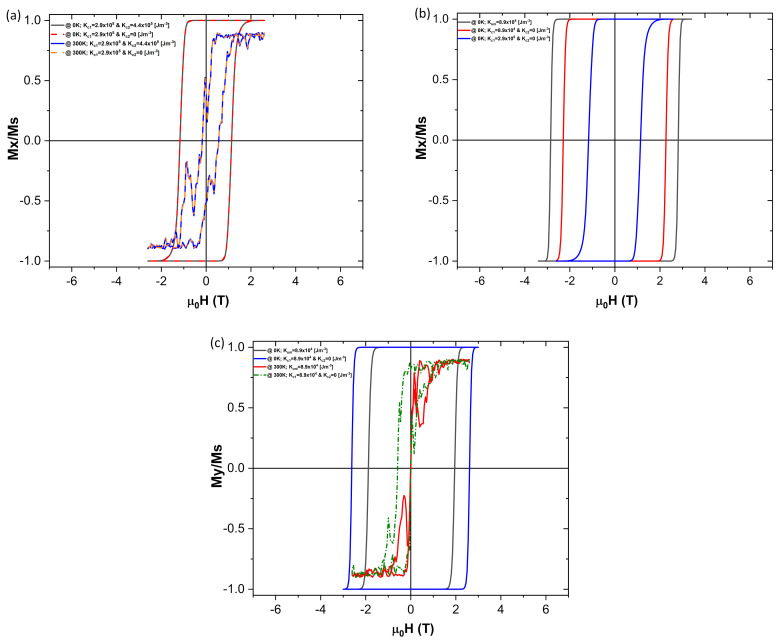
Comparison between cubic and uniaxial anisotropies. (**a**) Using first and second order cubic anisotropy constants, $K_{c1}$ and $K_{c2}$ respectively, where the easy anisotropy axis and $H_{\mathrm{ext}}$ are applied in the same v(1,0,0) direction. In both 0 K and 300 K there is no real benefit in using the second order term, $K_{c2}$. (**b**) For the constants values $K_{c1}=K_{uni}$, the simulation when just using the cubic anisotropy presents a coercive field ≈0.55 T lower than when just using the uniaxial one, being the $H_{\mathrm{ext}}$ applied along the same (1,0,0) anisotropy easy direction. The coercivity gets further reduced with a higher magnitude anisotropy constant. (**c**) When the external field, $H_{\mathrm{ext}}$, is applied along the (0,1,0) direction (which is perpendicular to the anisotropy one (1,0,0)), the coercive field for the cubic anisotropy is now ≈0.70 T higher than the uniaxial anisotropy one. This trend is seen both at 0 K and 300 K.

**Figure 8 nanomaterials-10-01149-f008:**
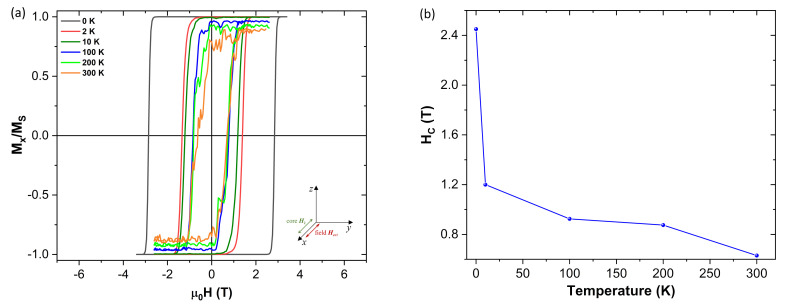
(**a**) Magnetic isotherm when applying the external magnetic field ($H_{\mathrm{ext}}$) parallel to the anisotropy axis ($H_{k}$ along the x-axis) of the NP (of core size 5.5 nm and 0.5 nm shell thickness). (**b**) Plot of the coercivity ($H_{c}$) versus temperature, showing how it rapidly decreases due to the presence of the thermal field.

**Figure 9 nanomaterials-10-01149-f009:**
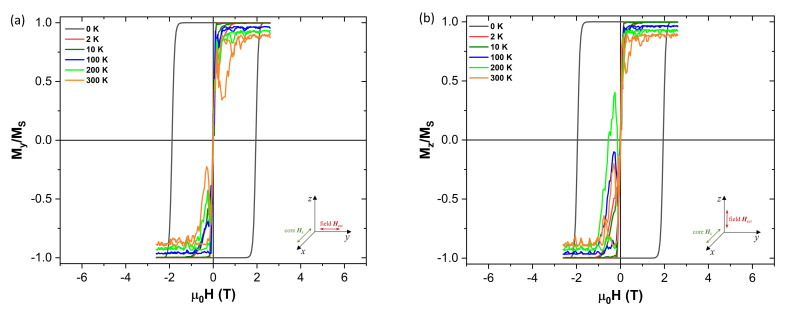
Magnetic isotherms when applying the external magnetic field ($H_{\mathrm{ext}}$) perpendicularly to the ordered core anisotropy direction ($H_{k}$ along the x-axis) of the NP (of core size 5.5 nm and 0.5 nm shell thickness). (**a**) When the field is applied along the y-axis and (**b**) when the field is applied along the z-axis. The coercivity rapidly tends to zero as soon as the thermal field is added, when the external magnetic field is applied perpendicularly to the anisotropy easy axis.

**Figure 10 nanomaterials-10-01149-f010:**
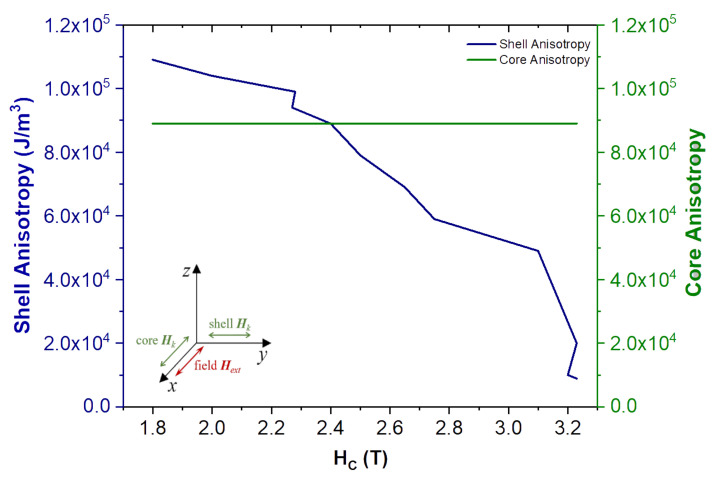
Effect of different anisotropy magnitudes and directions on the NPs core and shell. Anisotropy easy axis directions were defined as; (1,0,0) x-axis for the core; and (0,1,0) y-axis for the shell. As one can see, as the anisotropy in the shell is increased the coercivity of the entire NP decreases, pointing to the importance of the anisotropy direction and magnitude, in core-shell structures, to the overall magnetization dynamics of the NP.

**Table 1 nanomaterials-10-01149-t001:** Basic magnetic parameters of the prototype magnetic NP, determined from the ZFC-FC curves and magnetization isotherms. Coercive field at 10 K, $H_{c}^{10}$; anisotropy field at 10 K $H_{k}^{10}$; saturation magnetization at 10 K and 300 K, $M_{S}^{10}$, $M_{S}^{300}$, remnant magnetization at 10 K, $M_{r}^{10}$; effective anisotropy constant $K_{\mathrm{eff}}$.

$H_{c}^{10}\left(T\right)$	$H_{k}^{10}\left(T\right)$	$M_{S}^{10}\left({\mathrm{Am}}^{2}/\mathrm{kg}\right)$	$M_{S}^{300}\left({\mathrm{Am}}^{2}/\mathrm{kg}\right)$	$M_{r}^{10}\left({\mathrm{Am}}^{2}/\mathrm{kg}\right)$	$K_{\mathrm{eff}}\left(J/m^{3}\right)$
1.28	4.2	97	73	53	$8.9\times 10^{4}$

**Table 2 nanomaterials-10-01149-t002:** Simulation parameters used on mumax3 for the core-shell NP with uniaxial anisotropy. $M_{S}$ is the saturation magnetization, $A_{ex}$ is the exchange energy, $K_{u}$ is the anisotropy constant with its direction vector, v and the damping parameter α.

	$M_{S}$ (A/m)	$A_{\mathrm{ex}}$ (J/m)	$K_{u}$ (J/m${}^{3}$), v(i,j,k)	α
Core	$7.74\times 10^{5}$	$1.50\times 10^{-11}$	$8.9\times 10^{4};\;$v(1,0,0)	0.01
Shell	$6.19\times 10^{5}$	$1.20\times 10^{-11}$	0	0.005

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
