# Peer review of "Understanding Magnetization Dynamics of a Magnetic Nanoparticle with a Disordered Shell Using Micromagnetic Simulations"

_nanomaterials, 2020, doi:10.3390/nano10061149_

Round 1

Reviewer 1 Report

This manuscript proposes a micromagnetic model of individual magnetic nanoparticles with a typical core/shell magnetic structure. The authors try to show the influence of different stimulation parameters on the resulting M(H) loop. The core/shell model is around for tens of years, applied to any kind of material, but recently there are good examples of SANS studies where they can map the spin structure inside particles. Ferrites like magnetite have a clear tendency to form disordered surface structure, surrounding a conventional ordered core region. On the other hand, cobalt ferrite shows a canted structure homogeneously on the entire volume. In this sense, their choice of the model material is not the best. See the following papers,

[1] K. L. Krycka, J. A. Borchers, R. A. Booth, Y. Ijiri, K. Hasz, J. J. Rhyne, S. A. Majetich, Phys. Rev. Lett. 2014, 113, 147203.

[2] K. Hasz, Y. Ijiri, K. L. Krycka, J. A. Borchers, R. A. Booth, S. Oberdick, S. A. Majetich, Phys. Rev. B 2014, 90, 180405.

The authors assume that the shell owns reduced parameters compared to the bulk. Why? Unless referring to particles of poor crystalline quality, with a structurally disordered shell, this can not be taken as a general assumption. Very often, one can see the opposite effect, with a large increment of the anisotropy constant and saturation magnetization due to the local surface structure. Ref. [1] above shows this effect experimentally. The deviations of nanoparticle systems from the bulk values are usually connected to the broken symmetry at the particle surface. For this reason, these local properties can be controlled by the use of organic coatings with dramatic effects, for instance, increasing or decreasing saturation magnetization above the bulk value. See, [3] M. Vasilakaki, N. Ntallis, N. Yaacoub, G. Muscas, D. Peddis, K. N. Trohidou, Nanoscale 2018, 10, 21244. On the other hand, the synthesis process itself, and subsequent prepossess can deeply characterize the effective surface properties of the final particles. See [4] K. Lee, S. Lee, B. Ahn, Chem. Mater. 2019, 31, 728.

This paper shows didactically how to select parameters looking for a convergence of results, but this is just the common practice, it is not of special value for a scientific publication. Simulations of M(H) loops like those in figure 5 should extend the field well above the point of saturation. This is not true, especially in panel c. Even if contributing with a small magnetic moment, the canted structures, especially on the surface, determines the overall configuration and how easily the magnetization is reversed, hence leading to incorrect estimation of Hc.

What kind of anisotropy has been implemented in the model?  The authors talk about a generic anisotropy axis, probably referring to a single easy one, and mention a perpendicular axis as a hard one. This is not the case since Cobalt ferrite owns cubic anisotropy with positive K1 and K2, hence an easy axis along the edge direction of a cube.

While the topic is interesting, especially for applications, the model is poorly implemented. It could give a qualitative description of some effects, but clearly, the wrong implementation of the correct anisotropy symmetry limits its value. The authors should reorganize this work more systematically and correct the incorrect underlying assumptions before considering the text for publication.

Author Response

We are very grateful for the insightful comments and suggestions provided by the reviewer. A point by point reply is given below and the revised manuscript in the "tracking mode" is also attached - please see the attachment.

The main conflict seems to be the definition of a magnetic core-shell model and the implementation of the real structure of NPs to the theoretical model. Therefore we revised the text to clearly state our motivation and explained the relevance of the approach used in our simulations. We also carefully studied the suggested references and we discuss them in the context of other relevant works. Finally, we carried out additional simulations to address the anisotropy issues claimed by the reviewer. Some more explanation to the comments of the reviewer is given below.

Addressing 1st paragraph points/remarks.

- “... Ferrites like magnetite have a clear tendency to form disordered surface structure, surrounding a conventional ordered core region. On the other hand, cobalt ferrite shows a canted structure homogeneously on the entire volume. In this sense, their choice of the model material is not the best. See the following papers, ”

[1] K. L. Krycka, J. A. Borchers, R. A. Booth, Y. Ijiri, K. Hasz, J. J. Rhyne, S. A. Majetich, Phys. Rev. Lett. 2014, 113, 147203.

[2] K. Hasz, Y. Ijiri, K. L. Krycka, J. A. Borchers, R. A. Booth, S. Oberdick, S. A. Majetich, Phys. Rev. B 2014, 90, 180405.

The reviewer claims that the model of a structurally and magnetically coherent core and disordered shell is not a general property of fine particles. The suggested references studied 11 nm CoFe2O4 NPs [1] and magnetite NPs [2], which means a different type of NPs than we considered. When inspecting the works published on CoFe2O4, one can find multiple cases of Coey-like magnetic core-shell architecture:

e.g. https://www.hindawi.com/journals/jnm/2013/741036/

https://onlinelibrary.wiley.com/doi/full/10.1002/ppsc.201900061

https://pubs.acs.org/doi/10.1021/jp8016634

https://ieeexplore.ieee.org/document/8918243

https://www.sciencedirect.com/science/article/pii/S0304885312002624?via%3Dihub

https://link.springer.com/article/10.1007%2Fs11051-013-1767-2

https://aip.scitation.org/doi/10.1063/1.126727

https://pubs.acs.org/doi/pdf/10.1021/cm203280y

https://aip.scitation.org/doi/pdf/10.1063/1.126727

https://pubs.acs.org/doi/full/10.1021/jp9912307 and many others.

It is a well-known fact that the fabrication procedures determine the structure, size (distribution), and thus the final magnetic properties. A lot of work still has to be done to have a good classification and standardization methods for NPs, despite the good advances done recently (doi:10.3390/ijms160920308, https://doi.org/10.1088/1361-6463/aa7fa5).

The methods based on thermal decomposition of iron/cobalt oleates are among the most popular methods nowadays, and the NP architecture is very similar for all of them for a specific NP size (and size distribution).

So even though the suggested references [1] and [2] do present very interesting work using SANS to map the spin structure of NPs, our simulated particle is much smaller and our prototype cobalt ferrite sample has different architecture (as addressed above) than the ones measured in those references.

The Coey-like model of our cobalt ferrite NP is also supported by recent work carried out by the group of Disch together with some of our collaborators. They studied spatially resolved, non-correlated surface spin disorder in isolated, spherical cobalt ferrite nanoparticles of the same size and prepared by the same method by a combination of polarized small-angle neutron scattering and micromagnetic analysis (https://arxiv.org/pdf/1912.04081.pdf or Ph.D. thesis of D. Zakutna, 2016). They point out that the field-dependence of the coherently magnetized particle volume is not accessible with conventional integral measurement techniques (typically the "magnetic size" from SANS is comparable to that obtained from the Langevin isotherm analysis, e.g. https://journals.iucr.org/j/issues/2007/s1/00/ks6006/ks6006.pdf). They clearly demonstrated the presence of intra-particle spin disorder, which is coherent with our model.

Addressing 2 nd  paragraph points/remarks

- “The authors assume that the shell owns reduced parameters compared to the bulk. Why? Unless referring to particles of poor crystalline quality, with a structurally disordered shell, this can not be taken as a general assumption. Very often, one can see the opposite effect, with a large increment of the anisotropy constant and saturation magnetization due to the local surface structure. Ref. [1] above shows this effect experimentally. The deviations of nanoparticle systems from the bulk values are usually connected to the broken symmetry at the particle surface. For this reason, these local properties can be controlled by the use of organic coatings with dramatic effects, for instance, increasing or decreasing saturation magnetization above the bulk value. See, [3] M. Vasilakaki, N. Ntallis, N. Yaacoub, G. Muscas, D. Peddis, K. N. Trohidou, Nanoscale 2018, 10, 21244. On the other hand, the synthesis process itself, and subsequent prepossess can deeply characterize the effective surface properties of the final particles. See [4] K. Lee, S. Lee, B. Ahn, Chem. Mater. 2019, 31, 728. ”

As it is stated by the reviewer and in the mentioned reference, the effective anisotropy of NPs can be very different from the bulk ones and it also significantly vary with temperature (http://dx.doi.org/10.1016/j.jmmm.2012.03.019). Therefore we adopted the experimental value as adequate one and the uniaxial anisotropy for the micromagnetic simulations.

For comparison, we completed our study by an example of varying the anisotropy of the shell and we also carried out additional simulations using the cubic anisotropy and compared the results to our own starting point one.

We would like to kindly point out that the statement “The deviations of nanoparticle systems from the bulk values are usually connected to the broken symmetry at the particle surface.” is not valid in general. It is true that the atoms at the surface do have unsaturated bonds, thus the symmetry of the coordination polyhedron (an octahedron, a tetrahedron in case of spinels) is lower, which for the Fe or Co ions means that the quenched orbital moment becomes non-zero and it can give rise to quite a significant contribution to the total angular momentum, hence the magnetic moment. However, this symmetry-driven surface effect is not the only effect present in the real NPs. More often (and it has been demonstrated many times), the  NPs have a certain STRUCTURAL disorder far below the surface and the shell responds like a softer magnet than the core (as also seen in https://arxiv.org/pdf/1912.04081.pdf) or it sometimes mimics paramagnetic or spin glass-like behavior. In both cases, the use of bulk anisotropy parameters is not suitable and not corroborated experimentally although it is definitely true that the bulk cobalt ferrite does have a cubic anisotropy with large and positive K1 and K2.

The coexistence of cubic and uniaxial anisotropy is observed, with a dominance of the uniaxial anisotropy for particle size below 5nm (Peddis et al, Chem. Mater. 2012, 24, 6, 1062–1071 or Moumen, N.; Bonville, P.; Pileni, M. P. J. Phys. Chem. 1996, 100, vol.34, 14410−14416). The same point is made in the ongoing paper (Chem. Mater. 2013, 25, 10, 2005–2013), which demonstrates how different synthesis approaches can influence the NP anisotropy arrangement. Pointing to the importance of the fabrication procedure, a recent paper (doi:10.1002/ppsc.201900061) presents that the density of defects and shape of the nanocrystal can promote variations in Ms and anisotropy. For example in the last sentences of section 2.3 the authors mention how the different synthesis processes can lead to either a cubic or uniaxial anisotropy. Therefore using the uniaxial anisotropy for cobalt ferrite NPs of a sufficiently small size is not unrealistic.

We also regret to conclude that in the cited works, although we value their suggestion, there is no deeper analysis of the STRUCTURAL disorder in the context of the spin structure. For example in “[4] K. Lee, S. Lee, B. Ahn, Chem. Mater. 2019, 31, 728.” there is no direct evidence of how the structural coherence of the shell influences the coherence of the spin structure in the shell. Surprisingly, the shifts in the blocking temperature are negligible (btw just a different level of liquid helium or simply precision of the magnet power supply will give a different magnitude of the external field, which can cause such as shift) and the magnetic moment/size and effective anisotropy distribution is not analyzed at all. Moreover, the NPs are cubic in shape and much larger in size therefore the total magnetic anisotropy is expected to be different to a Coey-like NP as the shape anisotropy will be important. In this case, we believe that using the bulk values and additional shape anisotropy term is a relevant model.

In the work “[3] M. Vasilakaki, N. Ntallis, N. Yaacoub, G. Muscas, D. Peddis, K. N. Trohidou, Nanoscale 2018, 10, 21244” the effect of ligand exchange is discussed. Although the DFT calculations are very detailed, the experimental analysis does not point unambiguously to the findings from the DFT. There is no information on how the ligand exchange influences the structure and chemical composition of the particles, especially at the surface (for example even a simple ligand exchange from oleic acid to citric acid causes strong Co depletion of the surface layer: https://doi.org/10.1155/2016/7091241). Unfortunately, the mean magnetic volume (or size) and distribution of the blocking temperature (energy barrier) is not determined. According to about 20 years of experience in the in-field Mössbauer spectroscopy, the very simple analysis reported therein is not sufficient to prove that the change of the coating is the only factor changing the final magnetic properties. The Mössbauer spectra (MS) are a complex superposition of contributions from all Fe ions across the particle and thus distribution function analysis of the MS parameters as a function of the magnetic field is needed to get a piece of reasonable information about the evolution of the structural disorder and spin structure (e.g. https://doi.org/10.1063/1.4881331). In this vein, in case of an NP of such a size, the relative contribution of the surface spins to the MS will be hardly addressable, namely without the sophisticated analysis. Therefore it is not clear whether the “surface effect” (unsaturated coordination environment giving rise to non-zero orbital momentum) dominates over the “disorder effect” (gradient of lattice parameters, bond length, vacancies, depletion of Co or Fe, etc.).

Addressing 3rd   paragraph points/remarks

“This paper shows didactically how to select parameters looking for a convergence of results, but this is just the common practice, it is not of special value for a scientific publication. Simulations of M(H) loops like those in figure 5 should extend the field well above the point of saturation. This is not true, especially in panel c. Even if contributing with a small magnetic moment, the canted structures, especially on the surface, determines the overall configuration and how easily the magnetization is reversed, hence leading to incorrect estimation of Hc.”

These simulations are time-consuming, in particular when using small time steps for higher numerical precision. The reason why it seems that in Figure 5 the simulations do not go into the saturation point is just because of the fact that the parameters were set to rationally reduce the computational time.

Nonetheless, looking attentively to the panels of Figure 5, all of the simulations have reached well enough to saturation. In particular, the ones with the thermal field made no sense to continue to simulate behind the point as shown since the oscillations present are coherent with the thermal field behavior. Of course with much higher thermal fields they would see themselves reduced and higher on the saturation level. For the simulated fields, even the canted shell saturates, especially in the case of smaller thicknesses.

We also believe that the information on optimization of the parameters is useful for the community as quite often in publications including micromagnetic simulations such optimization is not done (or not done in a robust way).

Addressing 4th   paragraph points/remarks

“What kind of anisotropy has been implemented in the model?  The authors talk about a generic anisotropy axis, probably referring to a single easy one, and mention a perpendicular axis as a hard one. This is not the case since Cobalt ferrite owns cubic anisotropy with positive K1 and K2, hence an easy axis along the edge direction of a cube. ”

We assume the uniaxial anisotropy as already explained in the text above. However, we also completed the study using bulk cobalt ferrite values. In order to clarify this point, we updated section D in the context of the references given above.

We mention applying the external magnetic field perpendicular to the hard uniaxial easy axis (1,0,0). We also updated the text of section D to make this point clear.

Reviewer 2 Report

The nanoparticles belong to the intensively studied magnetic systems, showing a high potential in diverse applications. In-deep studies of the physical nature of processes governing the behavior of NPs are certainly welcome and very valuable. The present paper falls into that category, reporting results of micromagnetic simulations performed on a core-shell type spherical nanoparticle with realistic initial parameters taken from the experiment performed by Authors. The study is aimed at capturing the mechanism of the magnetization reversal and at characterization of the influence of various physical parameters of the NP (like size or anisotropy energy) as well as on the importance of the purely computational parameters (time steps). The crucial importance of the presence or absence of the easy-axis anisotropy in the shell and its relative magnitude with respect to the anisotropy in the core was found.

The study is a very careful one, the influence of the software settings on the calculated physically relevant quantities was extensively discussed. The selected topic is interesting and of importance, the discussion of the results is clear, coherent, convincing, absorbing and keeps the attention of the Reader.

I certainly recommend the manuscript for publication in Nanomaterials journal.

Below I list some truly minor remarks for the Authors to consider (related to the details of presentation only):

  • In the abstract: “the shell thickness increases” might be replaced with “the shell thickness ranges from…to”.
  • Page 2, line 11 from top: there is some problem with reference number – “7?”.
  • Page 3, line 2 from top: I guess the Authors would emphasize more that they took the parameters from their own measurements.
  • Page 3, line 14: probably “where each cell has its own magnetization”.
  • Page 3: maybe it would be interesting to mention how many cubic cells were used for core and shell with the typical geometry (core/shell size) used in the study (of course it can be easily calculated, but it is somehow interesting).
  • Page 3, line 7 from bottom: maybe it would be more clear to enumerate the particular parameters whose values are taken as 20% lower in the shell.
  • Page 4, caption, table 2: “Where;” should be corrected. Also, it might be useful to mention in the caption that vector v is the direction of the anisotropy.
  • Page 4, line 3 of the section RESULTS AND DISCUSSION: maybe it would be more clear to state” material-dependent and material-independent computational parameters".
  • Page 5, fig. 3, caption: maybe it would be better to remove “actual” describing the simulation time.
  • Page 6, line 6 from top: “Given that most literature and research are done” might be rephrased.
  • Page 6, fig. 4, caption: maybe “as α is increased, the coercivity is lowered” might sound better.
  • Page 7, line 6 from top: probably there should be “since if the shell is thicker” or “since when the shell is thicker”.
  • Page 7, line 9: “even though”
  • Page 7, line 11 (and in other places): I guess that the term “first neighbor” used here might make an impression that this is a typical model with spin Hamiltonian on the lattice. To be more precise, maybe it might be better to use the term “nearest neighbor cells” or similar throughout the text.
  • Page 7, line 15: “less smooth”.
  • Page 7, fig. 5, caption: maybe “(d) directly compares…”.
  • Page 8, line 10: The expression exp(KV) should have a bracket.
  • Page 8, fig. 6, caption: maybe it would be better to write “with corresponding insets showing the spin structure during the transition process”.
  • Page 8, line 2 from bottom: “even if just slightly to 2 K.” could be rephrased.
  • Page 9: “magnified by the perpendicular to anisotropy applied external magnetic hysteresis field.” could be rephrased.
  • Page 10, line 5 from top: probably “this greatly affects”.
  • Page 11: at the very end of Acknowledgments there was some problem with a LaTeX command.
  • Supplementary materials, in many places: probably “results of the simulations”.

Author Response

We are very grateful for the insightful comments and suggestions provided by the reviewer, which helped us to improve our manuscript. A point by point reply is given below and the revised manuscript in the "tracking mode" is also attached  - please see the attachment.  

  • In the abstract: “the shell thickness increases” might be replaced with “the shell thickness ranges from…to”.

→ The change was made as suggested. 

  • Page 2, line 11 from top: there is some problem with reference number – “7?”.

→ Resolved. There was a Latex issue referring to the second reference at this point. 

  • Page 3, line 2 from top: I guess the Authors would emphasize more that they took the parameters from their own measurements.

→ That paragraph was rewritten to better emphasize that the experimental measurements and NPs were done in our own lab and they are in good agreement with other results on NPs obtained by the same method. 

  • Page 3, line 14: probably “where each cell has its own magnetization”.

→ Corrected.

 Page 3: maybe it would be interesting to mention how many cubic cells were used for core and shell with the typical geometry (core/shell size) used in the study (of course it can be easily calculated, but it is somehow interesting).

→ Added to the last paragraph on page 3.

  • Page 3, line 7 from bottom: maybe it would be more clear to enumerate the particular parameters whose values are taken as 20% lower in the shell.

→ Specified directly that the Ms and Aex are about 20% smaller on the text, plus added the reference to table II where those values are represented.

  • Page 4, caption, table 2: “Where;” should be corrected. Also, it might be useful to mention in the caption that vector v is the direction of the anisotropy.

→ Specified directly both points are addressed by removing the “Where” and adding the description of the vector v as the anisotropy direction vector. 

  • Page 4, line 3 of the section RESULTS AND DISCUSSION: maybe it would be more clear to state” material-dependent and material-independent computational parameters".

→ Rewritten the last part of the 1st paragraph to make that point clearer.

  • Page 5, fig. 3, caption: maybe it would be better to remove “actual” describing the simulation time.

→ Changed as suggested.

  • Page 6, line 6 from top: “Given that most literature and research are done” might be rephrased.

→ Rephrased full sentence to make it clearer.

  • Page 6, fig. 4, caption: maybe “as α is increased, the coercivity is lowered” might sound better.

→ Changed as suggested.

  • Page 7, line 6 from top: probably there should be “since if the shell is thicker” or “since when the shell is thicker”.

→ Changed accordingly to the first suggestion.

  • Page 7, line 9: “even though”

→ Corrected.

  • Page 7, line 11 (and in other places): I guess that the term “first neighbor” used here might make an impression that this is a typical model with spin Hamiltonian on the lattice. To be more precise, maybe it might be better to use the term “nearest neighbor cells” or similar throughout the text.

→ Changed throughout as suggested.

  • Page 7, line 15: “less smooth”.

→ Changed as suggested.

  • Page 7, fig. 5, caption: maybe “(d) directly compares…”.

→ Caption changed slightly in that part to make it clearer.

  • Page 8, line 10: The expression exp(KV) should have a bracket.

→ Changed as suggested.

  • Page 8, fig. 6, caption: maybe it would be better to write “with corresponding insets showing the spin structure during the transition process”.

→ Changed as suggested.

  • Page 8, line 2 from bottom: “even if just slightly to 2 K.” could be rephrased.

→ Rephrased as “even for a thermal field as week as the one given by the 2 K temperature

  • Page 9: “magnified by the perpendicular to anisotropy applied external magnetic hysteresis field.” could be rephrased.

→ Rephrased as  This effect is now amplified by the fact that the external field is now applied perpendicularly to the anisotropy easy axis.”

  • Page 10, line 5 from top: probably “this greatly affects”.

→ Rephrased as “...and thus it greatly affects the overall magnetization dynamics.

  • Page 11: at the very end of Acknowledgments there was some problem with a LaTeX command.

→ The problem with project reference in LaTeX was addressed.

  • Supplementary materials, in many places: probably “results of the simulations”.

→ We are sorry, but we are not sure about the suggested revision. We checked the SI again for typos and grammar issues.

Round 2

Reviewer 1 Report

I read the arguments and assertions of the authors in response to my comments. I would like to clarify a bit of the discussion. As the authors state that the Coey model with a disordered shell is the generally accepted one to qualitatively justify the observation of variation of properties of nanoparticles compared to bulk counterpart. Modeling nanoparticles with an extra shell with different proprieties from the bulk has proved to be an effective way to explain a long list of experimental observations, such as enhanced anisotropy compared to bulk. However, I wanted to underline the recent relevant observations that shed additional light on the phenomenon. The two works I have mentioned are among the very few having the unique value of considering only fully crystalline particles and to be among the really few to give a direct experimental determination of the spin configuration. This makes them good model systems to investigate the magnetic spin configuration independently of structural influences such as amorphous/disordered surface or internal defects, e.g., dislocations.

[1] K. L. Krycka, J. A. Borchers, R. A. Booth, Y. Ijiri, K. Hasz, J. J. Rhyne, S. A. Majetich, Phys. Rev. Lett. 2014, 113, 147203.

[2] K. Hasz, Y. Ijiri, K. L. Krycka, J. A. Borchers, R. A. Booth, S. Oberdick, S. A. Majetich, Phys. Rev. B 2014, 90, 180405.

The [1] deals with magnetite, showing excellent agreement with the presence of a pure magnetic core/shell model. On the other hand, the hard-cobalt ferrite sample in [2] exhibits a disorder that extends over the entire particle, which forms a homogeneous disordered phase. The fact that these particles are larger than those simulated should provide additional confidence in considering that the disorder at the surface causes a full reorganization of the particle even for smaller ones.

This discussion does not intend to diminish the value of the results obtained by the authors, since one cannot exclude that for Cobalt ferrite, due to all possible variables related to the preparation process, the pure magnetic core/shell could not be demonstrated. On the other hand, since it seems that the presence of cobalt favors the extended disordered configuration, I consider that the authors, selecting cobalt ferrite as a model system, must mention that this is not the only possible configuration.

At the same time, the authors should clarify if they intend to study purely magnetic core/shell structures or structural core/shells that are intrinsically carrying also a magnetic core/shell behavior. I have intended this paper as trying to address the case of pure magnetic core/shell but reading the answers provided, and especially https://arxiv.org/abs/1912.04081 it seems to me that the authors are interested in systems exhibiting structural AND magnetic disorder. I know the paper of D. Zàkutnà et al., this is an excellent study of both structural and magnetic properties. They carefully investigate cobalt ferrite NPs which exhibit a magnetic core/shell-like structure, but their model is specifically addressing samples with a clear structural disordered shell (and some internal dislocations too), while [2] simply deals with a crystalline homogeneous ordered sample.

About the LONG LIST of references proposed, the authors should be more careful. Several of the references do not even mention core/shell structures or surface effects, others confirm the observation of bulk-like properties thanks to organic coatings, most of them clearly state the presence of a structural disordered shell, and none of them study experimentally the real spin configuration. Actually, one of them even remarks the possibility of a disorder that extends over the entire particle.

The core/shell model can be the perfect one every-time a structural distinction is present and even in case of fully ordered samples like magnetite-maghemite, but recent experimental evidence suggests that it is not the only possible configuration, especially for cobalt ferrite. This is something to mention in the manuscript, and I would like to remark that this is not in contrast with the results of this article. Moreover, I understand the fact that most of the samples in literature easily show some degree of structural disorder, intrinsically matching the description of the core/shell model but this needs to be specified, and the authors need to be clearer in identifying the framework of the application of their model.

Author Response

We are very grateful for the additional insightful comments based on a careful inspection of the arguments and additional references we provided.

Despite the changes to the first version of the manuscript, it has been pointed out by the reviewer that the definition of “our” core-shell model is not clear. Therefore we revised the text to clearly define the meaning of the core-shell model in the framework of our micromagnetic study.

We also agree that the spin structures in nanoparticles are a very complex phenomenon and the spin structures vary across different spinel ferrites depending on the cation and preparation procedure. Therefore we also modified the Introduction to improve the current state of the art based on the references and suggestions given by the reviewer. In particular, the different arrangements in cobalt ferrite observed in some recent studies. We have to point out that in the framework of micromagnetic simulations, it is not possible to distinguish between the structurally disordered/amorphous shells as the origin of spin disorder or spin disorder due to the surface or size effects. The different response of the shell and the core can be modeled by incorporating different magnetic parameters for both components. Therefore our model is purely magnetic core-shell by definition, but it can be useful for simulation of magnetic properties of nanoparticles with the dominant structural disorder, which gives rise to spin disorder in the shell if relevant input parameters are used.  Finally, the reviewer is right that some of the added references do not discuss the spin disorder, however they all show (not all discussing properly) that the crystallinity of NPs is quite variable and it is important for the final spin structure. Nevertheless, the presented references were only included if they were relevant for other discussions, e.g. on magnetic anisotropy. The list of changes in the updated text is given below. The changes are trackable using the "trackchanges.sty" of the Latex file.

Abstract

We did minor modifications to the abstract to account for the changes in the main text (clear definition of the model and its relevance, existence of more possible spin structures).

Introduction

We extended the text by adding a paragraph mentioning that other spins structures are common in magnetic nanoparticles, e.g. a homogeneous disorder as suggested by the reviewer.

We also updated the part on the definition of our model, which is can be termed magnetic core/shell. We also added an explanation that the micromagnetic concept allows us to account for different magnetic parameters of the core and the shell, but it is not possible to implement whether the effects come from surface/size phenomenon or due to structural disorder in the shell.

After the revision, we also removed some sentences with identical information.

Results and Discussion, Part C.

We added more information on how the external magnetic field causes an overall suppression of the spin disorder in the shell. The simulations clearly show that the spins in the shell tend to align with the core's magnetization, as observed in the recent experiment (https://arxiv.org/abs/1912.04081). We also had a chance to briefly discuss the correspondence of our simulations and relevance of the model (in spite of not being capable of disentangling the structural and spin disorder) used with the first author of the experimental study.

Results and Discussion, Part D-2

 The role of varying the shell anisotropy has been emphasized, I particular the effect that the lower the shell anisotropy the easier for the shell spins to align in the magnetic field following the core reversal for the given model.

Conclusions

We added again the definition of the model used in the micromagnetic modeling and we explained again briefly the relevance of applying the uniaxial anisotropy. We also commented on the variability of the spin arrangements in nanoparticles and the relevance of the micromagnetic simulation for their description.